# RETHINKING ENTROPY INTERVENTIONS IN RLVR: AN ENTROPY CHANGE PERSPECTIVE

## ABSTRACT

While Reinforcement Learning with Verifiable Rewards (RLVR) can enhance LLM reasoning, its training process poses a critical risk: Entropy Collapse. This phenomenon is a rapid loss of policy diversity, stemming from the exploration-exploitation imbalance and leading to suboptimal solutions. Recent entropy-intervention methods aim to prevent this, yet their underlying mechanisms remain unclear. In this paper, we conduct extensive experiments to reveal token-level entropy changes and how existing entropy intervention methods help avoid entropy collapse. Our findings point out a fundamental limitation of existing methods: they attempt to control the entropy indirectly. By only adjusting related factors, such as the advantage signal and generation probability, their effectiveness is inherently limited and prone to failure. To address this limitation, we introduce an entropy-change-aware reweighting scheme, namely **S**tabilizing **T**oken-level **E**ntropy-chang**E** via **R**eweighting (**STEER**), that adaptively stabilizes entropy dynamics through fine-grained, token-level adjustments. This approach prevents over-exploitation while ensuring robust exploration. Our extensive experiments demonstrate that **STEER** significantly avoids entropy collapse, stabilizes entropy dynamics, and achieves stronger downstream performance across math reasoning benchmarks.

## 1 INTRODUCTION

The success of Reinforcement Learning with Verifiable Rewards (RLVR) in advancing LLM reasoning (Jaech et al., 2024; Lambert et al., 2024; Guo et al., 2025; Shao et al., 2024; Yang et al., 2025a; Team et al., 2025) is largely attributed to its ability to foster emergent capabilities like long-form chain-of-thought (CoT) and self-reflection (Shao et al., 2024; Zhu et al., 2025b). However, a key challenge in RLVR is the exploration-exploitation trade-off under outcome-based supervision (Yeo et al., 2025; Yue et al., 2025). This is because rewards based solely on the final answer can force models into a state of premature convergence, where models stick to narrow solutions and ignore other correct ones. This issue is particularly damaging to group-based policy-gradient methods (Shao et al., 2024; Ahmadian et al., 2024), as the lack of output diversity makes it difficult to estimate relative advantages, thus providing weak learning signals.

This lack of output diversity is a direct consequence of a poorly managed exploration-exploitation trade-off. Policy entropy is the primary metric used to quantify this balance (Wu et al., 2025; Song et al., 2025; Li et al., 2025; Cui et al., 2025b): low entropy indicates insufficient exploration (a state of over-exploitation), while high entropy indicates sufficient exploration. Therefore, preventing a catastrophic drop in this metric, known as the Entropy Collapse, becomes a central research focus in RLVR. To avoid the entropy collapse, existing approaches attempt to indirectly influence entropy dynamics through several mechanisms, each with inherent limitations. One strategy targets (i) PPO-style ratio-clipping thresholds, for example, by decoupling them to enhance exploration (Yu et al., 2025); however, this approach can induce asymmetric and uncontrolled effects on entropy change. Another focuses on (ii) the relative weighting of positive and negative samples, either by up-weighting rare-but-correct solutions (He et al., 2025) or skewing weights towards negative samples (Zhu et al., 2025a). While effective at preventing over-sharpening, this method only modulates entropy as a byproduct and lacks fine-grained control. The third approach involves (iii) an entropy-induced advantage (Cheng et al., 2025; Tan & Pan, 2025; Wang et al., 2025b;a; Deng et al., 2025). This design, however, often has an unintended negative effect; it tends to excessively focus learning

on high-entropy tokens, which, instead of stabilizing entropy, amplify its fluctuations and can distort the entropy change. These observations lead to an important question: Is there a unified framework that can not only explain the root cause of limitations of existing methods, but also guide us to design better solutions?

We believe the answer is to analyze the problem from the perspective of entropy dynamics. We argue that the overall entropy dynamics during training arise from the accumulation of per-token entropy changes; thus, analyzing entropy change at the token level helps reveal the entropy dynamics. In this paper, we unify the entropy-intervention in RLVR through the lens of entropy change: we conduct a quantitative analysis of token-level entropy change, which not only allows us to analyze interesting properties and limitations of existing entropy-intervention methods, but also motivates us to propose a simple yet effective method to control entropy change.

Specifically, we start by conducting a quantitative analysis under mild conditions. Based on this analysis, we conceptually explain how existing methods influence entropy dynamics: (i) PPO-style ratio-clipping thresholds induce asymmetric effects on entropy change; (ii) the relative weighting of positive and negative samples modulates entropy change; and (iii) entropy-induced-advantage approaches magnify entropy fluctuations, which potentially accelerate entropy decline. Although these methods can mediate influence entropy change, they fall short of controlling entropy change directly. Guided by this insight, we introduce an entropy-change-aware scheme, called **S**tabilizing **T**oken-level **E**ntropy-chang**E** via **R**eweighting (**STEER**), that provides fine-grained, token-level control of policy entropy dynamics to keep per-step entropy change within a moderate band. In this way, our method steers the policy away from over-exploitation and sustains adequate exploration. Empirically, our method achieves superior downstream performance over strong baselines while effectively preventing entropy collapse and strengthening exploration across RLVR benchmarks.

In summary, our contributions can be briefly summarized as follow:

- We propose a quantitative analysis framework for entropy change and the entropy effect of existing entropy interventions can be unified and elucidated through token-level analysis.
- To precisely stabilize entropy change, we propose an adaptive and fine-grained reweighting method that keeps per-step entropy change within a moderate band.
- Experiments on standard RLVR setups demonstrate superior performance, training stability, and precise control of entropy.

## 2 PRELIMINARIES

### 2.1 RLVR ALGORITHMS

Given a prompt $q$ sampled from data $\mathcal{D}$, $\pi$ is denoted as the policy parameterized with $\theta$, and $o$ is denoted as the response sampled from $\pi_{old}(\cdot|q)$. PPO (Schulman et al., 2017) optimizes the policy by maximizing the expected advantage and stabilizes the training process through the clipped surrogate. Instead of training an additional value model, GRPO (Shao et al., 2024) samples a group of rollouts $o_{i\,i=1}^{G}$ for each prompt $q$ and estimates advantages by relative rewards within the group:

$$A_{i,t} = \frac{R_i - \text{mean}(\{R_i\}_{i=1}^{G})}{\text{std}(\{R_i\}_{i=1}^{G})}, \tag{1}$$

where $R_i$ equals 1 when the response is correct and $-1$ when the response is wrong for all tokens in the $i$-th response. Formally, by adapting the token-level policy gradient loss (Yu et al., 2025), GRPO maximizes the following objective.

$$\mathcal{J}(\theta) = \mathbb{E}_{q\sim\mathcal{D},\,\{o_i\}_{i=1}^{G}\sim\pi_{old}(\cdot|q)} \left[ \frac{1}{\sum_{i=1}^{G}|o_i|} \sum_{i=1}^{G} \sum_{t=1}^{|o_i|} \min\left(r_{i,t}A_{i,t}, (r_{i,t}, 1+\varepsilon, 1-\varepsilon)A_{i,t}\right) \right], \tag{2}$$

where $r_{i,t} = \frac{\pi_\theta(o_{i,t}|q,o_{i,<t})}{\pi_{old}(o_{i,t}|q,o_{i,<t})}$ denotes the importance sampling ratio. The KL divergence term between the current policy $\pi_\theta$ and the reference policy $\pi_{\text{ref}}$ in the original form (Shao et al., 2024) is excluded in this work.

## 2.2 POLICY ENTROPY OF LLMS

Entropy quantifies the uncertainty of a policy model's action selection under a given state (Haarnoja et al., 2018). The token entropy on token $o_{i,t}$ is defined as the Shannon entropy of the conditional distribution $\pi_\theta(\cdot|q, o_{i,<t})$:

$$\mathcal{H}_{i,t} = -\mathbb{E}_{o_{i,t} \sim \pi_\theta(\cdot|q, o_{i,<t})} \left[ \log \pi_\theta(o_{i,t}|q, o_{i,<t}) \right]. \tag{3}$$

Policy entropy measures a policy model's overall uncertainty on a dataset by averaging token entropy over sequences and positions. For policy model $\pi_\theta$ on dataset $\mathcal{D}$ the policy entropy is defined as:

$$\mathcal{H}(\pi_\theta, \mathcal{D}) = \mathbb{E}_{q \sim \mathcal{D}, \{o_i\}_{i=1}^G \sim \pi_{old}(\cdot|q)} \frac{1}{\sum_{i=1}^G |o_i|} \sum_{i=1}^G \sum_{t=1}^{|o_i|} \mathcal{H}_{i,t}. \tag{4}$$

## 3 ENTROPY-INTERVENTION MECHANISM: AN ENTROPY CHANGE PERSPECTIVE

Policy entropy serves as an indicator of a model's output diversity. The overall entropy change reflects the exploration–exploitation trade-off during training. Macro changes in policy entropy arise from the accumulation of micro entropy changes, with a single update's effect on a single token's conditional entropy constituting the atomic unit. In this section, we begin from this micro-level perspective, deriving a quantitative analysis to identify the direct factors that govern token-level entropy change. We then leverage this analysis to examine the impact of existing training parameters on the overall entropy dynamics.

### 3.1 QUANTITATIVE ANALYSIS ON TOKEN-LEVEL ENTROPY CHANGE

We start by analyzing the factors that govern a single token's entropy change. The policy gradient of GRPO (in Eq.(2)) is expressed as follows:

$$\nabla_\theta J(\theta) = \mathbb{E}_{q \sim \mathcal{D}, \{o_i\} \sim \pi_{old}(\cdot|q)} \left[ \frac{1}{\sum_{i=1}^G |o_i|} \sum_{i=1}^G \sum_{t=1}^{|o_i|} \mathbb{I}_{clip} r_{i,t} A_{i,t} \nabla_\theta \log \pi_\theta(o_{i,t} \mid q, o_{i,<t}) \right], \tag{5}$$

where

$$\mathbb{I}_{clip} = \begin{cases} 0, & A_{i,t} > 0 \text{ and } r_{i,t} > 1 + \varepsilon_{high}, \\ 0, & A_{i,t} < 0 \text{ and } r_{i,t} < 1 - \varepsilon_{low}, \\ 1, & \text{otherwise.} \end{cases} \tag{6}$$

During the RLVR training process, token-level logit distributions are influenced by multiple factors, so it is impractical to estimate the induced entropy change in entropy directly. To capture the essence of distribution shifts, we follow the assumption from (Liu, 2025):

**Assumption 1** (Parameter-independent softmax). *For any context (state) $s_{i,t} = (q, o_{i,<t})$, each token (action) $a$ in vocabulary $\mathcal{V}$ is associated with an independent logit parameter $z_{s,a}(\theta)$. And the next-token distribution follows $\pi_\theta^k(\cdot \mid s) = \text{softmax}(z_{s,\cdot}^k)$.*

Assumption 1 states that a gradient step on the sampled token does not substantially affect the logits of the other tokens in the vocabulary. Given this assumption, we obtain the following theorem (see proof in Appendix C).

**Theorem 1.** *(First–order entropy change) Let policy model $\pi_\theta$ follow Assumption 1. The change of conditional entropy between two update steps is defined as $\Delta\mathcal{H}_{it} \triangleq \mathcal{H}(\pi_\theta^{k+1} \mid s_{i,t}) - \mathcal{H}(\pi_\theta^k \mid s_{i,t})$. Then the first-order estimation of $\Delta\mathcal{H}_{it}$ in Eq. 2 is*

$$\Omega_{i,t} = -\eta \, \mathbb{E}_{a \sim \pi_\theta^k(\cdot|s_{i,t})} \, w_{i,t}(1 - \pi_\theta^k(a|s_{i,t}))^2 \left( \log \pi_\theta^k(a|s_{i,t}) + \mathcal{H}(\pi_\theta^k \mid s_{i,t}) \right), \tag{7}$$

*where $\eta$ is the learning rate, $w_{i,t} = \mathbb{I}_\varepsilon \, r_{i,t} \, A_{i,t}$ is per-token weight.*

Theorem 1 above implies that, under Assumption 1, the entropy change of a single token $\Delta\mathcal{H}_{it}$ can be reflected by $\Omega_{i,t}$. Obviously, $\Omega_{i,t}$ are jointly determined by learning rate $\eta$, per-token gradient weight $w_{i,t}$, generation probability $\pi_\theta^k(a|s_{i,t})$ and current conditional entropy $\mathcal{H}(\pi_\theta^k \mid s_{i,t})$.

In contrast to our milder Assumption 1, prior work often relies on more restrictive assumptions to derive entropy change. For instance, (Cui et al., 2025b) (denoted as *Cov*) assumes a uniform entropy distribution across different queries within the same batch. However, this assumption is often unrealistic and can lead to estimations that misrepresent the ground-truth entropy dynamics.

To validate our approach, we compare our entropy change estimator, $\Omega_{i,t}$, with that of *Cov* during a standard GRPO training process. As visualized in Figure 1, our proposed $\Omega_{i,t}$ closely tracks the ground-truth entropy change, showing a positive correlation. While the estimation *Cov* shows only a weak correlation.

To quantify this gap, we compute the Mean Squared Error (*MSE*), Pearson Correlation Coefficient (*PCC*), and Spearman's Rank Correlation Coefficient (*SRCC*) between each estimation and the ground-truth token-level entropy change, as shown in Figure 2. Across all three metrics, $\Omega_{i,t}$ from Theorem 1 delivers orders-of-magnitude lower MSE and sub-

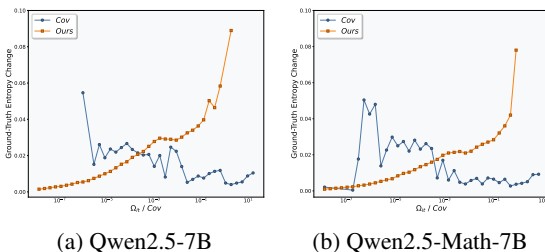

(a) Qwen2.5-7B      (b) Qwen2.5-Math-7B

Figure 1: Entropy change estimation in the first 10 training steps on Qwen2.5-Math-7B and Qwen2.5-7B. The curve compares estimated vs. ground-truth entropy change (left axis) and histograms show token counts per bin (right axis).

stantially higher *PCC* and *SRCC* than *Cov*. Furthermore, the SRCC between $\Omega_{i,t}$ and the ground-truth token entropy change exceeds $60\%$ across all models, demonstrating a strong rank correlation. A more comprehensive comparison is provided in Appendix E.3. These results strongly validate the effectiveness of our estimator derived in Theorem 1 and the soundness of Assumption 1.

| *Model* | *Method* | *MSE* ↓ | *PCC* ↑ | *SRCC* ↑ |
|---|---|---|---|---|
| *Math-1.5B* | *Cov* | 5.37 | -6e-5 | +0.04 |
|  | *Ours* | 5e-4 | +0.42 | +0.65 |
| *Qwen-7B* | *Cov* | 0.53 | +0.05 | +0.08 |
|  | *Ours* | 8e-4 | +0.39 | +0.72 |
| *Math-7B* | *Cov* | 0.29 | +0.03 | +0.06 |
|  | *Ours* | 4e-4 | +0.42 | +0.61 |

Figure 2: MSE, PCC and SRCC comparison.

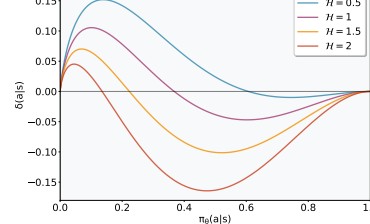

Figure 3: Token-level entropy change indicator $\delta(a|s)$.

## 3.2 ON ANALYSIS OF PHENOMENA IN ENTROPY DYNAMICS

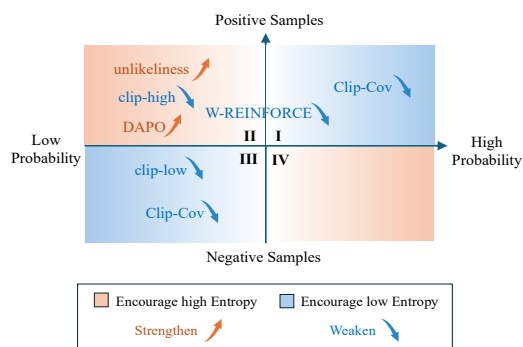

| **Method** | $\pi_\theta(a|s)$ | $A(a|s)$ | $\mathcal{H}(\cdot|s)$ |
|---|---|---|---|
| DAPO | ✓ | ✓ | ✗ |
| Unlikeliness | ✓ | ✓ | ✗ |
| W-REINFORCE | ✗ | ✓ | ✗ |
| Entropy Adv. | ✗ | ✓ | ✓ |
| KL Reg. | ✓ | ✗ | ✗ |
| Entropy Reg. | ✗ | ✗ | ✓ |
| Forking Tokens | ✗ | ✗ | ✓ |
| Clip-Cov | ✓ | ✓ | ✗ |
| **STEER** | ✓ | ✓ | ✓ |

Figure 4: Entropy change with advantage and probability.

Figure 5: Key Considerations in Current Approaches.

### 3.2.1 ENTROPY DYNAMICS UNDER ADVANTAGE AND PROBABILITY

To dissect the factors governing token-level entropy change, we first need to decompose the first-order estimation $\Omega_{i,t}$ from Theorem 1. To this end, we introduce a **token-level entropy change indicator**, $\delta(a|s)$, defined as:

$$\delta(a|s) = -\pi_\theta(a|s)(1 - \pi_\theta(a|s))^2(\log(\pi_\theta(a|s)) + \mathcal{H}(\cdot|s)) \tag{8}$$

This allows us to express the entropy change from Theorem 1 as $\Omega_{i,t} = \eta \, \mathbb{E}_{a \sim \pi_\theta(\cdot|s_{i,t})}[w'_{i,t} \cdot \delta(a|s_{i,t})]$, where $w'_{i,t}$ contains the magnitude-scaling terms like advantage $A(a|s_{i,t})$ and the importance sampling ratio. The key insight is that $\delta(a|s)$ represents the *intrinsic directional tendency* of the entropy change, since it only depends on the token's generation probability $\pi_\theta(a|s)$ and the current conditional entropy $\mathcal{H}(\cdot|s)$. Figure 3 visualizes $\delta(a|s)$ as a function of these two variables.

Based on this decomposition, we can now analyze the entropy dynamics by examining how token-level entropy changes with different signs of the advantage $A(a|s)$ and the indicator $\delta(a|s)$. To illustrate, we create a two-dimensional space, shown in Figure 4, which can be divided into four distinct quadrants:

**Quadrant I: Exploitation (Entropy Decrease).** For high-probability correct tokens ($A > 0, \delta < 0$), rewarding an already-mastered behavior concentrates probability mass, thus *decreasing* entropy.

**Quadrant II: Exploration (Entropy Increase).** For low-probability correct tokens ($A > 0, \delta > 0$), rewarding a rare-but-correct behavior diversifies the policy, thereby *increasing* entropy.

**Quadrant III: Suppression (Entropy Decrease).** For low-probability incorrect tokens ($A < 0, \delta > 0$), penalizing an unlikely error pushes its probability further toward zero, which also *decreases* entropy.

**Quadrant IV: Error-Correction (Entropy Increase).** For high-probability incorrect tokens ($A < 0, \delta < 0$), penalizing an overconfident error flattens the distribution to encourage seeking alternatives, substantially *increasing* entropy.

To validate these theoretical findings, we conduct an experiment to provide empirical support. Specifically, we can learn from the above analyses that entropy increases in two of these quadrants: (Quadrant II) when updating on low-probability tokens with positive advantages, and (Quadrant IV) when updating on high-probability tokens with negative advantages. To test this, we selectively apply double-weighting (to strengthen) or masking (to weaken) to 10% of tokens falling into each quadrant and track the resulting entropy. As shown in Figure 6, all four interventions successfully increase policy entropy compared to the standard GRPO baseline, confirming our model's validity. Further experimental details are available in Appendix E.1.

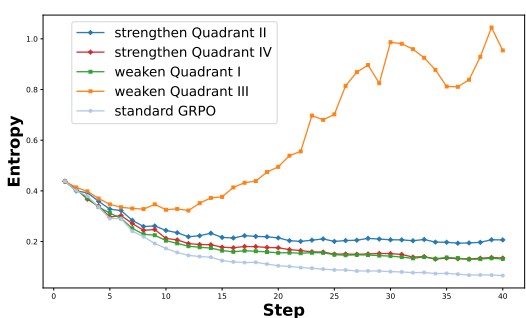

Figure 6: Four schemes to uplift entropy based on advantage and probability.

In a standard RLVR process, these four dynamics co-exist, acting as competing forces that shape the policy. Policy entropy evolves from the superposition of these updates. Consequently, **entropy collapse** can be understood as a state where the exploitation-driven, entropy-decreasing updates (Quadrants I and III) consistently overwhelm the exploration-driven, entropy-increasing updates (Quadrants II and IV). This framework not only explains the phenomenon but also provides a foundation for analyzing the effects of other interventions, such as positive/negative sample rebalancing and ratio clipping.

### 3.2.2 EXPLAINING THE ASYMMETRIC IMPACT OF RATIO CLIPPING

Ratio clipping is a core component of PPO-style algorithms, designed to prevent destructive policy updates by constraining the importance sampling ratio $r_t$. This mechanism can be interpreted within

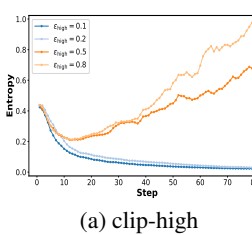 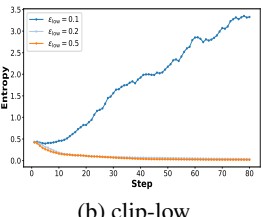 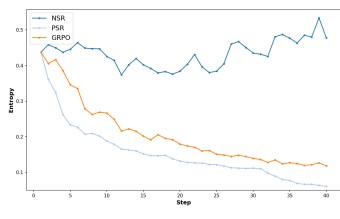

(a) clip-high      (b) clip-low

Figure 7: Entropy dynamics with ratio clipping.      Figure 8: PSR-NSR

our framework as a gate that primarily suppresses updates for tokens with large ratios—namely, low-probability tokens. As our analysis in Section 3.2.1 shows, these tokens correspond to the entropy-increasing Quadrant II (exploration) and the entropy-decreasing Quadrant III (suppression).

This insight allows us to form a clear hypothesis about how adjusting the clipping thresholds, $\varepsilon_{\text{high}}$ and $\varepsilon_{\text{low}}$, will asymmetrically affect policy entropy:

**Adjusting $\varepsilon_{\text{high}}$**: This threshold gates updates on positive-reward tokens. Increasing $\varepsilon_{\text{high}}$ (as in DAPO, (Yu et al., 2025)) relaxes the constraint on Quadrant II updates. This should unleash more of the natural, entropy-increasing effect of exploration. We therefore predict that **a higher $\varepsilon_{\text{high}}$ will increase policy entropy**.

**Adjusting $\varepsilon_{\text{low}}$**: This threshold gates updates on negative-reward tokens. Increasing $\varepsilon_{\text{low}}$ relaxes the constraint on Quadrant III updates. This should amplify the natural, entropy-decreasing effect of suppression. We therefore predict that **a higher $\varepsilon_{\text{low}}$ will decrease policy entropy**.

To verify our predictions, we conducted two experiments. First, we confirmed that clipping is indeed concentrated on low-probability tokens, as shown by the trigger counts in Figure 9. Second, we independently varied $\varepsilon_{\text{high}}$ and $\varepsilon_{\text{low}}$ and tracked the resulting entropy dynamics. The results, presented in Figures 7a and 7b, perfectly align with our predictions: entropy rises with a higher $\varepsilon_{\text{high}}$ and falls with a higher $\varepsilon_{\text{low}}$.

This analysis demonstrates that our framework provides a principled explanation for the asymmetric and often counter-intuitive effects of ratio clipping on policy entropy. The mechanism is not simply about limiting updates, but about selectively suppressing the competing forces of exploration and suppression that originate in the low-probability regions of the policy distribution.

### 3.2.3 EXPLAINING THE IMPACT OF POSITIVE AND NEGATIVE SAMPLE WEIGHTING

A notable phenomenon, observed by Zhu et al. (2025a) and confirmed in our experiments (Figure 8), is that training exclusively on negative samples (Negative Sample Reweighting, or NSR) sustains high policy entropy, whereas training only on positive samples (Positive Sample Reweighting, or PSR) leads to a rapid entropy collapse. Our four-quadrant framework provides a clear explanation for this behavior.

The key insight is that training data is naturally dominated by high-probability tokens. While our analysis shows that the *magnitude* of entropy change, $|\delta(a|s)|$, is similar for both high- and low-probability tokens (Figure 3), the sheer volume of high-probability tokens means they dictate the overall entropy trend.

**In PSR (Positive Sample Reweighting)**, the training signal is dominated by high-probability correct tokens, which fall into Quadrant I (Exploitation). This leads to a relentless decrease in entropy. Crucially, PSR removes all negative samples, thereby eliminating the powerful, entropy-increasing force of Quadrant IV (Error-Correction). Without this countervailing force, the policy quickly converges to a narrow solution set, causing entropy to collapse.

**In NSR (Negative Sample Reweighting)**, the training signal is dominated by high-probability incorrect tokens, which fall into Quadrant IV (Error-Correction). This provides a strong and continuous entropy-increasing signal. By removing all positive samples, NSR also eliminates the primary source of entropy decrease from Quadrant I (Exploitation). The result is a policy that constantly seeks to correct its errors, thereby maintaining high diversity and high entropy.

This framework also clarifies the mechanism behind other related methods. For instance, the strategy of up-weighting rare-but-correct tokens, as proposed by He et al. (2025) and Deng et al. (2025),

can be understood as a targeted intervention to boost the entropy-increasing effect of Quadrant II (Exploration). By amplifying this specific signal, these methods aim to counteract the dominant entropy-decreasing pressure from Quadrant I and thus mitigate entropy collapse.

### 3.2.4 THE PERILS OF TARGETING HIGH-ENTROPY TOKENS

While advantage and token probability determine the *direction* of an entropy update, the current conditional entropy, $\mathcal{H}(\cdot|s)$, governs its *magnitude*. Our analysis of the entropy change indicator $\delta(a|s)$ reveals a critical dynamic: the magnitude of potential entropy change, $|\delta(a|s)|$, increases significantly as $\mathcal{H}(\cdot|s)$ grows, particularly for high-probability tokens (Figure 3, right half). This implies that tokens in states of high uncertainty are inherently volatile and prone to large swings in entropy. This relationship is empirically confirmed in Figure 18, which shows a strong correlation between a token's current entropy and the magnitude of its subsequent entropy change.

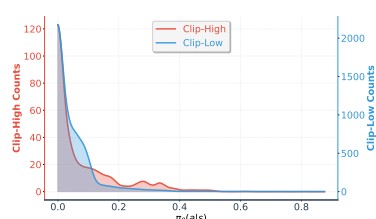

Figure 9: The average clip counts over 10 steps.

This volatility has led some methods, such as Entropy-based Advantage (Cheng et al., 2025) and GTPO (Tan & Pan, 2025), to propose interventions that explicitly up-weight high-entropy tokens. The intuition is that focusing on these uncertain states will promote exploration and thus increase overall policy entropy.

However, our analysis reveals this strategy to be counterproductive and potentially harmful. High-entropy tokens are not a reliable source of entropy *increase*; they are a source of entropy *variance*. By amplifying updates on these tokens, these methods create a dangerous positive feedback loop: When policy entropy happens to decrease, the amplified updates on the now lower-entropy (but still volatile) tokens can cause it to decrease even faster; This creates a system that is highly sensitive to its own fluctuations. Instead of stabilizing entropy, it amplifies its inherent oscillations.

We demonstrate this destabilizing effect in Figure 11. Compared to the standard GRPO baseline, entropy-induced advantage methods exhibit much larger fluctuations. Critically, when the policy enters a phase of decline, these methods can **accelerate entropy collapse**, leading to a faster and more severe drop in diversity. This finding highlights a key flaw in targeting high-entropy tokens: rather than preventing collapse, such interventions can inadvertently aggravate it.

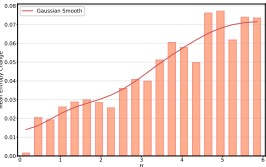

Figure 10: Empirical corelation between entropy and entropy.

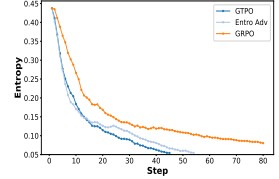

(a) Math-7B on DAPO-17k

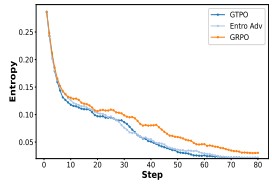

(b) Math-7B on Math

Figure 11: Entropy dynamics with ratio clipping.

## 4 STABILIZING TOKEN-LEVEL ENTROPY-CHANGE VIA RE-WEIGHTING

Building on the above analysis, we find that all three factors materially shape entropy change, whereas existing approaches target only a subset, which limits their effectiveness, as shown in Table 5. Since excessive entropy change can cause the policy entropy to rapidly increase or decrease, potentially leading to model training failure, we aim to keep the stepwise entropy change within a moderate range. To control entropy change precisely, we introduce an adaptive and fine-grained token-reweighting scheme that keeps the stepwise entropy change within a moderate band. Since $\Omega_{i,t}$ in Figure 2 shows a strong correlation with the ground-truth entropy change, a simple approach is to design a token-level weight negatively correlated with $\Omega_{i,t}$ to suppress updates of tokens with excessively large entropy changes.

Specifically, we apply an exponential-decay mapping to the token weights:

$$\lambda_{it} = e^{-k \cdot |\Omega_{i,t}|}, \text{ where } k = \frac{-\ln \lambda_{\min}}{\max\{|\Omega_{i,t}| \mid o_i \in \mathcal{B}\}}, \tag{9}$$

so that the token with the largest entropy change in each mini-batch attains the minimum weight. $\lambda_{\min}$ is the only hyperparameter introduced and enforces token weights within $[\lambda_{\min}, 1]$. $\lambda_{\min}$ equals to 1, STEER degenerates into standard GRPO. It is noteworthy that this reweighting scheme does not fundamentally hinder the model's learning, as the weighting is dominated by a few tokens with very large $\Omega_{i,t}$ within the batch, while the majority of tokens still have weights approaching 1.

## 5 EXPERIMENT

### 5.1 RLVR TRAINING SETUPS

**Training:** We conduct experiments on three different models, including Qwen2.5-Math-7B, Qwen2.5-Math-1.5B and Qwen2.5-14B. We adapt our training codebase from verl (Sheng et al., 2025) and follow the training recipe of standard GRPO. Our training data is DAPO-Math-17k (Yu et al., 2025), containing only math problems with integer ground-truth answers. Both the KL-divergence and entropy loss terms are removed in our experiments. Generation batch size is set to 512, and update batch size is set to 32. Rollout times are set to 8. Training is performed with top-p value of 1.0 and temperature= 1.0. Training details of our method and baselines are in Appendix D.

**Evulation:** We evaluated our models and baselines on six widely used mathematical reasoning benchmarks: AIME24, AIME25, AMC23 (Li et al., 2024), MATH-500 (Hendrycks et al., 2021), Minerva Math (Lewkowycz et al., 2022), and OlympiadBench (He et al., 2024), detailed in Appendix D. Validation is performed with a top-p value of 0.7 and temperature= 1.0 across all models and test sets. We use Math-Verify for training validation and final evaluation.

**Baselines:** For throughout comparison, we compare our method against 10 baselines, including standard GRPO (Shao et al., 2024), SimpleRL-Zoo (Zeng et al., 2025), Eurus-PRIME(Cui et al., 2025a), OPO (Hao et al., 2025), GRPO with cilp-high (Yu et al., 2025), GRPO with entropy loss (Schulman et al., 2017), GRPO with Fork Tokens (Wang et al., 2025b), W-REINFORCE (Zhu et al., 2025a), Entro. Adv. (Cheng et al., 2025), Clip-Cov and KL-Cov (Cui et al., 2025b).

**Main Results:** As shown in Table 1, STEER outperforms classical RLVR baselines as well as existing entropy intervention baselines across all datasets. STEER improves average performance by 2.7 over the runner-up (OPO) and by 3.4 over the runner-up (Clip-Cov) in the Entropy Intervention Baselines. The performance experiments on Qwen2.5-Math-1.5B and Qwen2.5-14B shown in Figure 4 are compared with the top3 competitors in Table 1 (i.e., OPO, Clip Cov, and Entro. Adv.). We also assessed the sensitivity of the experimental results to hyperparameters $\lambda_{\min}$, as shown in the Figure 15. It is evident that our method performs consistently well when $\lambda_{\min} \in [0.5, 0.8]$.

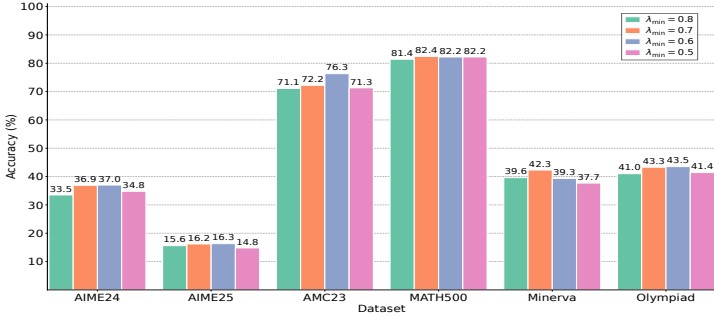

Figure 12: Advantage and Probability

Figure 13 shows the test curves during training, where STEER outperforms the baselines. Figure 14 presents the test curves for different hyperparameters, demonstrating both stability and superiority.
**Entropy Control** The superiority of our method is not only reflected in its performance but also in its ability to regulate entropy across a wide range. We consider an extreme training setup with

Table 1: Benchmark results of different methods (values are multiplied by 100; Avg. is the mean across six datasets).

| Method | AIME24 | AIME25 | AMC23 | MATH500 | Minerva | Olympiad | Avg. |
|---|---|---|---|---|---|---|---|
| Qwen2.5-Math-7B | 13.8 | 5.3 | 44.6 | 39.6 | 9.9 | 13.8 | 21.2 |
| **Classical RLVR Baselines** | | | | | | | |
| GRPO | 28.0 | 14.3 | 66.2 | 78.6 | 37.3 | 40.9 | 44.2 |
| SimpleRL-Zoo | 25.2 | 13.4 | 70.6 | 78.6 | 37.8 | 38.4 | 44.0 |
| Eurus-PRIME | 20.9 | 13.0 | 65.2 | 79.8 | 37.4 | 40.6 | 42.8 |
| OPO | 32.2 | 13.4 | 71.5 | 82.2 | 38.2 | 41.0 | 46.4 |
| **Entropy Intervention Baselines** | | | | | | | |
| GRPO w/ clip-high | 31.7 | 12.8 | 66.8 | 79.0 | 38.6 | 39.3 | 44.7 |
| GRPO w/ Entro. Loss | 29.1 | 14.0 | 67.6 | 80.0 | 38.2 | 37.9 | 44.5 |
| GRPO w/ Fork Tokens | 31.9 | 14.3 | 65.5 | 79.2 | 37.1 | 40.9 | 44.8 |
| W-REINFORCE | 31.9 | 14.3 | 65.5 | 79.2 | 37.1 | 40.9 | 44.8 |
| Entro. Adv. | 27.5 | 13.5 | 70.2 | 79.6 | 36.8 | 42.8 | 45.1 |
| Clip–Cov | 32.5 | 12.9 | 68.4 | 78.0 | 40.8 | 41.3 | 45.7 |
| KL–Cov | 32.8 | 14.1 | 64.2 | 78.8 | 37.1 | 39.4 | 44.4 |
| **Our Method** | | | | | | | |
| STEER | **36.9** | **16.2** | **72.2** | **82.4** | **42.3** | **43.3** | **49.1** |

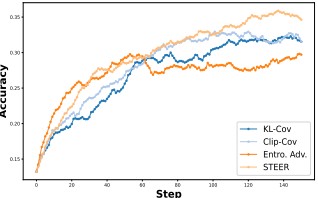 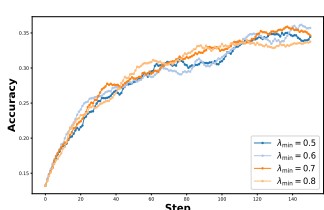

Figure 13: Test set accuracy dynamics comparison with different $\lambda_{\min}$

Figure 14: Test set accuracy dynamics comparison with benchmarks

$\varepsilon_{\text{high}} = 5$ and $\varepsilon_{\text{low}} = 0.99$, where almost no ratio clipping is applied. In such scenarios, RL training is vulnerable due to the influence of extreme values. The results are shown in the figure below:

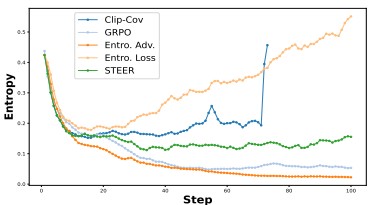 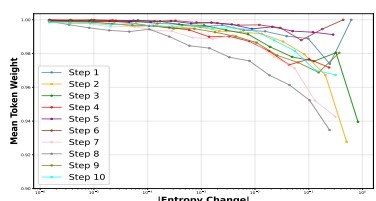

Figure 15: Entropy in extreme scenarios.

Figure 16: Advantage and Probability

# 6 CONCLUSION

In this paper, we rethink the entropy interventions through the lens of entropy change. By proposing a quantitative analysis framework for entropy change, the entropy effect of current entropy interventions can be unified and elucidated through token-level analysis. Motivated by stabilizing entropy change, we propose STEER, an adaptive, fine-grained reweighting scheme that precisely keeps per-step entropy changes within a moderate band by suppressing potentially disruptive updates. Extensive experiments on mathematical reasoning benchmarks demonstrate that STEER achieves superior performance, enhanced training stability. Our work provides both a new lens for analyzing RL dynamics and a practical solution for developing robust and effective training algorithms for LLMs.

## ETHICS STATEMENT

We have manually reevaluated the dataset we created to ensure it is free of any potential for discrimination, human rights violations, bias, exploitation, and any other ethical concerns.

## REPRODUCIBILITY STATEMENT

To ensure the reproducibility of our findings, all source code and datasets used in our experiments are included in the supplementary material. The provided materials are sufficient to replicate the main results presented in this paper.

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

## A  USAGE OF LLMs

Throughout the preparation of this manuscript, Large Language Models (LLMs) were utilized as a writing and editing tool. Specifically, we employed LLMs to improve the clarity and readability of the text, refine sentence structures, and correct grammatical errors. All final content, including the core scientific claims, experimental design, and conclusions, was conceived and written by us, and we take full responsibility for the final version of this paper.

## B  RELATED WORK

Entropy regularization (Mnih et al., 2016; Haarnoja et al., 2018), an early line of applied in traditional RL, may mislead actions at critical states (Zhang et al., 2025) and has been shown to be highly coefficient-sensitive in LLM training (Cheng et al., 2025; Cui et al., 2025b). The KL divergence term between the current policy $\pi_\theta$ and the reference policy $\pi_{\text{ref}}$ in the original form (Shao et al., 2024) is excluded, since its practical impact is often negligible or counterproductive for reasoning tasks, as demonstrated in recent works (Yu et al., 2025; Chu et al., 2025; Hu et al., 2025). (Liu et al., 2025) argues that the KL penalty not only preserves entropy but also acts as a regularizer, ensuring that the online policy remains close to a stable reference, which stabilizes learning and reduces overfitting to misleading reward signals. One typical approach to address entropy collapse is by raising the sampling temperature during inference. However, (Luo et al., 2025) findings suggest that while this method postpones the onset of entropy collapse, it does not prevent it, as entropy continues to decrease progressively throughout the training process. Recent studies have sought to mitigate entropy collapse by adjusting key elements of policy optimization, such as PPO-style ratio clipping (Yu et al., 2025; Yang et al., 2025b), balancing positive and negative samples (Zhu et al., 2025a), and applying KL regularization (Liu et al., 2025). However, these methods are broad and lack fine-grained control at the token level, with their mechanisms often not fully explained in a unified or principled way. To address this gap, researchers have increasingly used policy entropy as a critical measure for assessing the exploration-exploitation trade-off in RLVR (Wu et al., 2025; Song et al., 2025; Li et al., 2025). Policy entropy in LLMs has been widely recognized as a vital external indicator of this balance, with low entropy reflecting over-exploitation and insufficient exploration, while high entropy indicates the opposite. Although prior work (Cui et al., 2025b) considers entropy change, the resulting estimation is distorted (see Figure 1) due to its unreasonable state-equivalence assumption. Notably, its entropy-control scheme (i) enforces a hard binary split by entropy change without considering their intra-group differentiation, and (ii) may hinder learning process, since high-entropy-change tokens that are informative for exploration are over-penalized. the proposed entropy control method has two main limitations: (i) it imposes a hard binary partition of tokens by entropy change, with no intra-group granularity; (ii) it over-suppresses the contribution of high entropy-change tokens—often the most informative—thereby hindering learning. Further, the factors shaping entropy dynamics remain largely uncharacterized, constraining actionable control.

## C  THEOREM PROOF DETAILS

**Theorem 1.** *(First–order entropy change) Let policy model $\pi_\theta$ follows Assumption 1. The change of conditional entropy between two update steps is defined as $\Delta\mathcal{H}_{it} \triangleq \mathcal{H}(\pi_\theta^{k+1} \mid s_{i,t}) - \mathcal{H}(\pi_\theta^k \mid s_{i,t})$. Then the first-order estimation of $\Delta\mathcal{H}_{it}$ in Eq. 2 is*

$$\Omega_{i,t} = -\eta \, \mathbb{E}_{a \sim \pi_\theta^k(\cdot \mid s_{i,t})} \, w_{i,t}(1 - \pi_\theta^k(a \mid s_{i,t}))^2 \, (\log \pi_\theta^k(a \mid s_{i,t}) + \mathcal{H}(\pi_\theta^k \mid s_{i,t})), \tag{10}$$

*where $\eta$ is the learning rate, $w_{i,t} = \mathbb{I}_\varepsilon \, r_{i,t} \, A_{i,t}$ is per-token weight.*

*Proof.* The proof is similar to (Liu, 2025). Taking the first-order Taylor expansion, we have

$$\begin{aligned}
\Delta\mathcal{H}_{it} &\triangleq \mathcal{H}(\pi_\theta^{k+1} \mid s_{i,t}) - \mathcal{H}(\pi_\theta^k \mid s_{i,t}) \\
&\approx \left\langle \nabla_\theta \mathcal{H}\left(\pi_\theta^k \mid s_{i,t}\right), z^{k+1} - z^k \right\rangle.
\end{aligned}$$

Since we have log trick $\mathbb{E}_{a \sim \pi_\theta(\cdot|s)}[\nabla_\theta \log \pi_\theta(a \mid s)] = 0$, the gradient term can be derived as

$$
\begin{aligned}
\nabla_\theta \mathcal{H}(\pi_\theta \mid s) &= \nabla_\theta \mathcal{H}(\pi_\theta(\cdot \mid s)) \\
&= \nabla_\theta \big( - \mathbb{E}_{a \sim \pi_\theta(\cdot|s)} \big[ \log \pi_\theta(a \mid s) \big] \big) \\
&= - \mathbb{E}_{a \sim \pi_\theta(\cdot|s)} [\nabla_\theta \log \pi_\theta(a \mid s) + \log \pi_\theta(a \mid s) \, \nabla_\theta \log \pi_\theta(a \mid s)] \\
&= - \mathbb{E}_{a \sim \pi_\theta(\cdot|s)} [\log \pi_\theta(a \mid s) \, \nabla_\theta \log \pi_\theta(a \mid s)].
\end{aligned}
$$

Then we have

$$
\begin{aligned}
\Delta \mathcal{H}_{it} =& \big\langle \nabla_\theta \mathcal{H}(\theta^k \mid s_{i,t}), \, (z^{k+1} - z^k) \big\rangle \\
=& - \Big\langle \mathbb{E}_{a \sim \pi_\theta^k(\cdot|s_{i,t})} \big[ \log \pi_\theta(a \mid s_{i,t}) \nabla_\theta \log \pi_\theta(a \mid s_{i,t}) \big], \theta^{k+1} - \theta^k \Big\rangle \\
=& - \mathbb{E}_{a \sim \pi_\theta^k(\cdot|s_{i,t})} \Big[ \log \pi_\theta(a \mid s_{i,t}) \big\langle \nabla_\theta \log \pi_\theta(a \mid s_{i,t}), \theta^{k+1} - \theta^k \big\rangle \Big] \\
=& - \mathbb{E}_{a \sim \pi_\theta^k(\cdot|s_{i,t})} \Big[ \log \pi_\theta(a \mid s_{i,t}) \sum_{a' \in \mathcal{A}} \frac{\partial \log \pi_\theta(a \mid s_{i,t})}{\partial \theta_{s_{i,t},a'}} \big( \theta_{s_{i,t},a'}^{k+1} - \theta_{s_{i,t},a'}^k \big) \Big] \\
=& - \mathbb{E}_{a \sim \pi_\theta^k(\cdot|s_{i,t})} \Big[ \log \pi_\theta(a \mid s_{i,t}) \sum_{a' \in \mathcal{A}} \big( \mathbf{1}\{a = a'\} - \pi(a' \mid s_{i,t}) \big) \big( \theta_{s_{i,t},a'}^{k+1} - \theta_{s_{i,t},a'}^k \big) \Big] \\
=& - \mathbb{E}_{a \sim \pi_\theta^k(\cdot|s_{i,t})} \Big[ \big( \log \pi_\theta(a \mid s_{i,t}) - \mathbb{E}_{\hat{a} \sim \pi_\theta^k(\cdot|s_{i,t})} \log \pi_\theta(a \mid s_{i,t}) \big) \\
& \quad \big( \theta_{s_{i,t},a}^{k+1} - \theta_{s_{i,t},a}^k - \mathbb{E}_{a' \sim \pi_\theta^k(\cdot|s_{i,t})} \big( \theta_{s_{i,t},a'}^{k+1} - \theta_{s_{i,t},a'}^k \big) \big) \Big] \\
=& - \mathbb{E}_{a \sim \pi_\theta^k(\cdot|s)} \Big[ \log \pi_\theta^k(a|s) + \mathcal{H}(\cdot|s) \Big] \Big[ \big( 1 - \pi_\theta^k(\alpha|s) \big) \big( z_{s_{i,t},a}^{k+1} - z_{s_{i,t},a}^k \big) \Big] \\
=& - \mathbb{E}_{a \sim \pi_\theta^k(\cdot|s)} \Big[ \log \pi_\theta^k(a|s) + \mathcal{H}(\cdot|s) \Big] \Big[ w(s|a) \big( 1 - \pi_\theta^k(\alpha|s) \big)^2 \Big],
\end{aligned}
$$

where $w(s|a)$ is the weight of policy gradient. $\qquad \square$

# D  TRAINING SETTINGS

## D.1  DETAILED INFORMATION FOR TEST DATASET

Table 2: Dataset statistics.

| Test Datasets | #Questions | Level |
|---|---|---|
| AIME24 | 30 | Olympiad |
| AIME25 | 30 | Olympiad |
| AMC23 | 40 | Intermediate |
| MATH500 | 500 | Advanced |
| Minerva | 272 | Graduate |
| OlympiadBench | 675 | Olympiad |

## D.2  TRAINING DETAILS FOR OUR METHOD AND BASELINES.

All algorithms are implemented based on the official GRPO codebase within the VeRL framework. We use a learning rate of 1e-6 without warm-up across all experiments. At each rollout step, we generate 8 answers for each of 512 sampled questions, then split the data into 16 mini-batches and train the policy network for 16 gradient steps. Models are trained for at most 150 rollout steps.

Unless otherwise specified, we follow GRPO's default design choices with token-level loss normalization without dynamic sampling and KL regularization. For all models, the maximum input length is 1024 and the minimum input length is 3072. All the experiments were conducted on H20 GPUs.

# E  ADDITIONAL EXPERIMENTS

## E.1  STRENGTHS AND WEAKNESSES OF THE ENTROPY DYNAMICS

For experiments in Figure 6, we select samples with a generation probability greater than 0.8 and an advantage greater than 0, as well as those with a generation probability less than 0.2 and an advantage less than 0, and randomly mask 10% of them. Similarly, for samples with a generation probability greater than 0.8 and an advantage less than 0, or a generation probability less than 0.2 and an advantage greater than 0, we set the token weight of 10% to twice the original value.

Adjusting the proportion of enhancement or suppression also significantly impacts the change in entropy as follow:

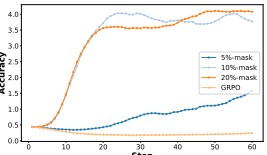 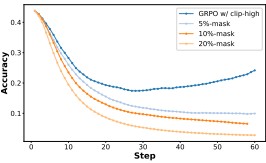

Figure 17: Empirical corelation between entropy and entropy.  Figure 18: Empirical corelation between entropy and entropy.

## E.2  HYPERPARAMETER ANALYSIS

STEER also consistently achieves the highest average performance on both Qwen2.5-Math-1.5B (38.1) and Qwen2.5-14B (45.1), demonstrating its superior capabilities in improving model reasoning.

Table 3: Hyperparameter Analysis

| Model | AIME24 | AIME25 | AMC23 | MATH500 | Minerva | Olympiad |
|---|---|---|---|---|---|---|
| $\lambda_{\min} = 0.8$ | 33.5 | 15.6 | 71.1 | 81.4 | 39.6 | 41.0 |
| $\lambda_{\min} = 0.7$ | 36.9 | 16.2 | 72.2 | 82.4 | 42.3 | 43.3 |
| $\lambda_{\min} = 0.6$ | 37.0 | 16.3 | 76.3 | 82.2 | 39.3 | 43.5 |
| $\lambda_{\min} = 0.5$ | 34.8 | 14.8 | 71.3 | 82.2 | 37.7 | 41.4 |

## E.3  ENTROPY CHANGE ESTIMATION COMPARISON

We recorded the token entropy changes for the first ten steps across different models and datasets, as shown in Figure 20 and21. It can be observed that our method exhibits a clear positive correlation, which strongly supports our theoretical framework.

## E.4  A TOKEN-LEVEL GRADIENT REWEIGHTING FRAMEWORK FOR SHAPING POLICY ENTROPY

In our analysis, existing entropy intervention methods can be unified into a gradient reweighting framework and subsequently examine their respective impacts on policy entropy. The table below summarizes the different weighting schemes used by existing methods, while our proposed approach is more fundamental, weighting based on entropy change.

Let $w_{i,t}(q) = \mathbb{I}(\pi_\theta, A_{i,t}) r_{i,t} \mathcal{A}(\pi_\theta, A_{i,t}) + \beta \mathcal{R}(\pi_\theta)$.

Table 4: Benchmark results of different models (example caption).

| Model | AIME24 | AIME25 | AMC23 | MATH500 | Minerva | Olympiad | Avg. |
|---|---|---|---|---|---|---|---|
| Qwen2.5-Math-1.5B | | | | | | | |
| base | 4.1 | 2.1 | 24.7 | 29.0 | 9.2 | 20.5 | 14.9 |
| GRPO | 16.2 | 7.6 | 56.0 | 74.4 | 26.1 | 34.6 | 35.8 |
| OPO | 14.8 | 9.0 | 58.2 | 72.2 | 26.1 | 35.9 | 36.0 |
| Entro. Adv. | 15.0 | 9.1 | 55.7 | 70.2 | 26.8 | 34.9 | 35.3 |
| Clip-Cov | 14.7 | 8.4 | 56.0 | 72.8 | 26.4 | 34.9 | 35.5 |
| STEER | 17.2 | 9.7 | 61.3 | 75.4 | 28.0 | 36.9 | 38.1 |
| Qwen2.5-14B | | | | | | | |
| base | 3.9 | 2.6 | 25.8 | 52.6 | 15.4 | 23.0 | 20.6 |
| GRPO | 17.2 | 13.2 | 66.3 | 80.6 | 38.0 | 42.2 | 42.9 |
| OPO | 17.8 | 12.6 | 68.2 | 78.6 | 37.7 | 42.6 | 42.9 |
| Entro. Adv. | 14.6 | 9.8 | 65.6 | 78.8 | 36.5 | 40.9 | 41.0 |
| Clip-Cov | 14.1 | 13.6 | 59.8 | 78.2 | 38.6 | 43.2 | 41.2 |
| STEER | 19.3 | 14.0 | 70.3 | 81.6 | 39.1 | 46.3 | 45.1 |

Figure 19: Entropy Change Estimation curves on DAPO-Math-17k.

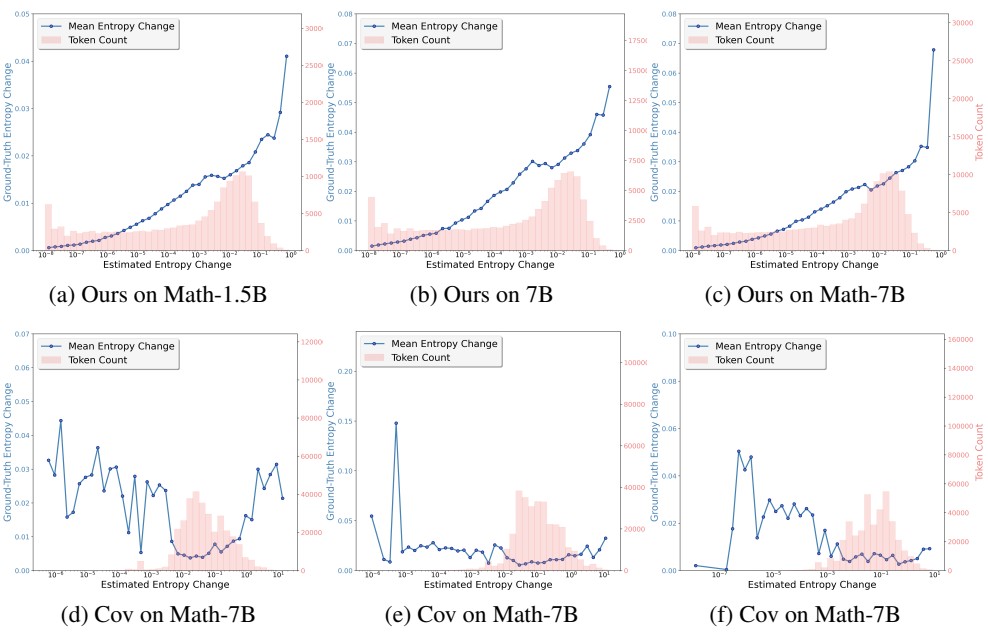

(a) Ours on Math-1.5B     (b) Ours on 7B     (c) Ours on Math-7B

(d) Cov on Math-7B     (e) Cov on Math-7B     (f) Cov on Math-7B

Figure 20: Entropy Change Estimation on DAPO-Math-17k.

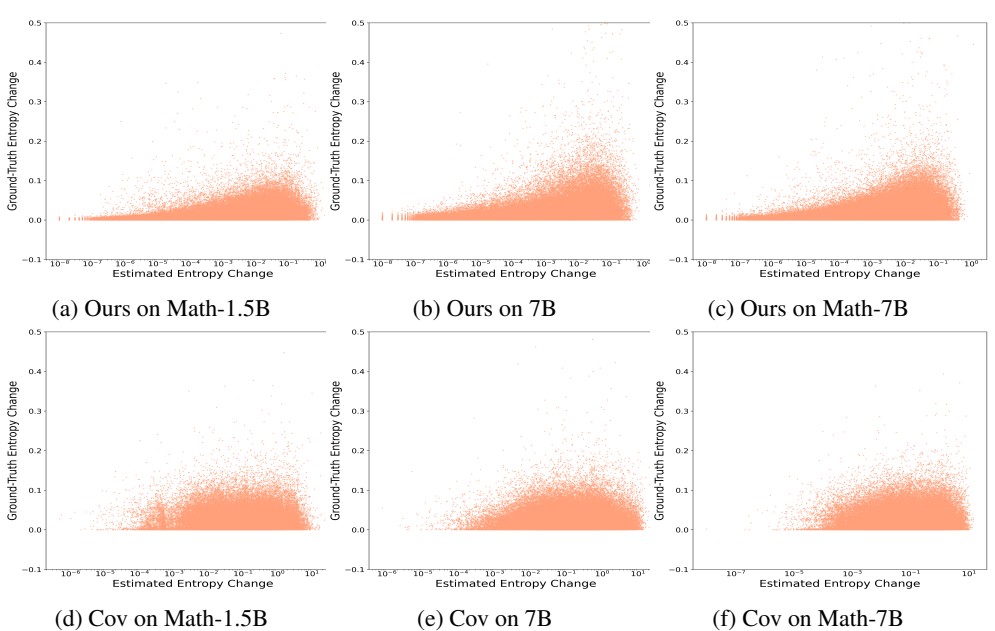

Figure 21: Entropy Change Estimation scatters on DAPO-Math-17k.

Table 5: A token-level Gradient Reweighting Framework for shaping policy entropy.

| Method | $w_{i,t}$ |
|---|---|
| DAPO / DCPO | $\mathbb{I}_\varepsilon \to \mathbb{I}_{\varepsilon_{\text{high}}, \varepsilon_{\text{low}}}$ |
| KL penalty | $\mathcal{R}\left(\pi_\theta\right) = \frac{\pi_{\text{ref}}(o_{i,t}\mid q, o_{i,<t})}{\pi_\theta(o_{i,t}\mid q, o_{i,<t})} - 1$ |
| Entropy Loss | $\mathcal{R}\left(\pi_\theta\right) = -\log \pi_\theta(o_{i,t}\mid q, o_{i,<t}) - 1$ |
| Unlikeliness | $\hat{R}_{i,t} = R_i\left(1 - \beta_{\text{rank}} \frac{G - \text{rank}(o_i)}{G}\right)$ |
| W-REINFORCE | $\mathcal{A}\left(\pi_\theta, A_{i,t}\right) = \begin{cases} \lambda, & A_{i,t} > 0 \\ 1, & A_{i,t} < 0 \end{cases} \quad \lambda < 1$ |
| Entropy Advantage | $\mathcal{A}\left(\pi_\theta, A_{i,t}\right) = A_{i,t} + \min\left(\alpha \cdot \mathcal{H}_{it}^{\text{detach}}, \frac{|A_{it}|}{\kappa}\right) \quad \alpha > 0, \kappa > 1$ |
| PPL-based | $\mathcal{A}\left(\pi_\theta, A_{i,t}\right) = A_{i,t}(1 - \alpha\log\text{-PPL}(o_i))$ |
| Position-based | $\mathcal{A}\left(\pi_\theta, A_{i,t}\right) = A_{i,t} + \gamma\text{sign}(A_{i,t})\sigma(r_{it}) \quad r_{it}$: token's relative position |
| Forking Tokens | $\mathbb{I}_\varepsilon = \mathbb{I}_\varepsilon \wedge \mathbb{I}(\mathcal{H}_{it} > \tau_\mathcal{D})$ |
| Clip-Cov | $\mathbb{I}_\varepsilon = \mathbb{I}_\varepsilon \wedge \mathbb{I}\left(\left(\log \pi_\theta(o_{i,t}) - \frac{1}{N}\sum_{j=1}^N \log \pi_\theta(y_j)\right)\left(A(o_{i,t}) - \frac{1}{N}\sum_{j=1}^N A(y_j)\right) > \tau_\mathcal{D}\right)$ |
| KL-Cov | $\mathcal{R}\left(\pi_\theta\right) = \frac{\pi_{\text{old}}(o_{i,t}\mid q, o_{i,<t})}{\pi_\theta(o_{i,t}\mid q, o_{i,<t})} - 1$ |

