# OpenReview forum: "Rethinking Entropy Interventions in RLVR:  An Entropy Change Perspective"
_ICLR.cc/2026/Conference — ICLR 2026 Conference Withdrawn Submission_

### Official Review · Reviewer_UYSj · 2025-10-28

**Soundness:** 3
**Presentation:** 2
**Contribution:** 3
**Rating:** 4
**Confidence:** 4

**Summary:**

The author gives a framework to reveal the underlying mechanisms of token-level entropy change.
The key idea is to quantitatively analyze the \delta_H(first order expansion) between single RL update and get every token's contribution.
Based on this framework, this paper gives the analysis of current entropy intervention methods.
Then the author gives a method which softly adjust the weights according to the token-level entropy change indicator.
The proposed method gets superior performance on math reasoning benchmark.

**Strengths:**

1. The analysis framework is clear. Based on it, the author provides unified entropy interventions analysis for current methods.

2. The idea hidden in the entropy change mechanisms is very interesting, which can provide insight for how to do entropy intervention for other researchers.

**Weaknesses:**

The paper presents an interesting and potentially important idea, but it requires substantially more explanation and clarification to strengthen its contributions.

## Major Concerns

1. The description of the four distinct quadrants (lines 231–242) lacks clarity. According to Figure 3, if a token receives a high probability, its corresponding \(\Delta\) indicator appears to be negative. If this interpretation is correct, please provide a clear and explicit explanation of this relationship, including any underlying assumptions or derivations.

2. As illustrated in Figure 6, all interventions increase policy entropy compared to GRPO. However, it is unclear why the "weaken Quadrant III" option results in an unstable entropy increase. Could the authors elaborate on the underlying mechanism? Additional analysis or theoretical justification would greatly improve understanding here.

3. The method section (Section 4) is overly concise and lacks sufficient detail. The discussed entropy change mechanism does not appear to be directly tied to the proposed method. The authors should more explicitly explain how insights from the entropy change analysis motivate or derive the proposed approach. Furthermore, it would be valuable to discuss whether more sophisticated or fine-grained techniques could be employed to refine the method.

## Minor Issues and Other Questions

1. There appears to be an error in line 103, where the clipping function (e.g., \(\texttt{Clip}\)) may have been omitted. Additionally, if \(\epsilon_{\text{high}}\) and \(\epsilon_{\text{low}}\) are introduced here, please define and explain them explicitly at their first mention to avoid confusion.

2. The histogram referenced in the caption or text of Figure 1 is missing from the figure itself. Please include it or clarify its intended location.

3. Figure 5 includes a table comparing methods but lacks citations for each entry. Please add appropriate references to ensure reproducibility and proper attribution.

4. In line 376, the reference to "Figure 2" seems incorrect. Moreover, Figure 19 in the appendix is not visible or properly rendered. Please verify and correct these figure references and ensure all appended figures are legible.

**Questions:**

See Weaknesses

---

### Official Review · Reviewer_QEaB · 2025-11-01

**Soundness:** 3
**Presentation:** 1
**Contribution:** 3
**Rating:** 4
**Confidence:** 3

**Summary:**

The key of this paper is to rethink the entropy intervention methods in Reinforcement Learning with Verifiable Rewards (RLVR) from the perspective of entropy change. It proposes a new framework to understand and improve existing methods. The paper introduces a new entropy-change-aware reweighting scheme and conducts extensive experiments on mathematical reasoning benchmarks, demonstrating the new method's superiority in avoiding entropy collapse, stabilizing entropy dynamics, and enhancing downstream performance.

**Strengths:**

- **Originality**: The paper re-examines the entropy intervention issue in RLVR from the perspective of entropy change, revealing the limitations of existing methods and proposing new solutions. This provides a novel approach to understanding and improving RLVR.
- **Quality**: The authors validate the effectiveness of the new method through extensive experiments and compare it with various existing methods. The experimental results show that the new method achieves certain performance improvements across multiple benchmarks.
- **Clarity**: The structure of the article needs improvement. The captions for Figures 13 and 14 are reversed, and Figures 15 and 16 are not explained. The logic needs enhancement, and the captions for each figure are insufficient. The clarity of the paper needs improvement.

**Weaknesses:**

1. Although the paper proposes a quantitative analysis framework to study entropy change, it is based on some assumptions: each token's logit parameters are independent. These assumptions may not hold in practical applications, which could affect the accuracy and applicability of the theoretical analysis. The authors could further discuss the rationality of these assumptions and validate their impact in experiments.
2. The paper mainly focuses on the entropy intervention issue in RLVR and validates the proposed method on mathematical reasoning tasks. The effectiveness and applicability of the method need more thorough validation. The authors could consider testing the new method in a wider range of scenarios to prove its universality.
3. In the experimental section, although the authors provide detailed experimental settings and results, they do not elaborate on the selection basis and adjustment process of the hyperparameter $\lambda_{min}$ in the reweighting scheme. The authors could provide more details about parameter selection and adjustment.
4. In Section 3.2.1, why are high-probability correct tokens classified as A > 0 and $\delta$ < 0? How is this classification determined?
5. Why does increasing $\epsilon_{high}$ lead to an increase in entropy? It seems counterintuitive that relaxing the clip constraint would result in a more deterministic policy, which should decrease entropy.
6. Is reweighting necessary? If the impact is significant, wouldn't clipping $\Omega$ off also reduce the impact on tokens?

**Questions:**

See Weaknesses.

---

### Official Review · Reviewer_FL9t · 2025-11-01

**Soundness:** 2
**Presentation:** 2
**Contribution:** 2
**Rating:** 2
**Confidence:** 4

**Summary:**

This paper investigates the problem of "Entropy Collapse" in Reinforcement Learning with Verifiable Rewards (RLVR) for training Large Language Models (LLMs). The authors' main contribution is the introduction of the first-order estimator for token-level entropy change, then they proposes STEER (Stabilizing Token-level Entropy-changE via Reweighting). STEER aims to directly control entropy by applying a reweighting scheme that suppresses updates for tokens with a large magnitude of estimated entropy change, thereby keep the training stability. Experiments trained on Qwen2.5-Math-7B show that STEER outperforms standard GRPO and other entropy-intervention baselines, achieving higher accuracy (Table 1) and demonstrating more stable entropy dynamics during training (Figure 15).

**Strengths:**

The paper's core contribution utilizes a first-order estimator for token-level entropy change, which, in my opinion, is novel.

**Weaknesses:**

- The paper seems to imply that maintaining a stable, horizontal entropy line is superior, as suggested by Figure 15. It is questionable whether a flat entropy curve is always desirable. Neither related literature nor existing experimental results conclusively demonstrate that stabilizing entropy at a constant level is inherently beneficial.

- The paper's results lack credibility. The experiments are limited to a single model, Qwen2.5-Math-7B. Furthermore, the reported performance gain over the baseline is excessively large (an average of +4.9% over GRPO and +3.4% over the runner-up baseline, Clip-Cov). Based on my extensive experience with math RL training, an average improvement exceeding 2% across six math benchmarks *solely* from an algorithmic change is already substantial. The claimed improvement raises suspicions of cherry-picking. I recommend that the paper validate the generalizability of its method on at least three different training datasets or three different base models.

- The entropy dynamics vary significantly across different models. For instance, popular open-source models like R1-Distill-Qwen, Qwen3, Qwen2.5, and the Llama series all exhibit different entropy curves during training. Given that the paper's main contribution revolves around the analysis of entropy *change*, I believe it is essential to include analyses on at least three models from different model families.

- Some of the paper's analyses, such as the one in Figure 4 and Figure 7, is not original. The conclusions presented are already implicitly suggested in prior work like DAPO (Yu et al., 2025) and Cui et al. (2025b) . Unless the authors can provide more rigorous theoretical results, I find the contribution in this area to be insufficient.

- Equation 9 is difficult to understand and lacks intuitive explanation . The paper should elaborate on the intuition behind this formulation.

**Questions:**

See weakness

---

### Note · Authors · 2025-11-30

I have read and agree with the venue's withdrawal policy on behalf of myself and my co-authors.